# SH3BP2 Deficiency Ameliorates Murine Systemic Lupus Erythematosus

**DOI:** 10.3390/ijms22084169

**Published:** 2021-04-17

**Authors:** Kyoko Kawahara, Tomoyuki Mukai, Masanori Iseki, Akiko Nagasu, Hajime Nagasu, Takahiko Akagi, Shoko Tsuji, Sumie Hiramatsu-Asano, Yasuyoshi Ueki, Katsuhiko Ishihara, Naoki Kashihara, Yoshitaka Morita

**Affiliations:** 1Department of Rheumatology, Kawasaki Medical School, Kurashiki 701-0192, Japan; kyoko.k0925@gmail.com (K.K.); yoshida5655@gmail.com (A.N.); akagitahiko@gmail.com (T.A.); shoko7.05.13@gmail.com (S.T.); h061eb@gmail.com (S.H.-A.); morita@med.kawasaki-m.ac.jp (Y.M.); 2Department of Immunology and Molecular Genetics, Kawasaki Medical School, Kurashiki 701-0192, Japan; miseki@med.kawasaki-m.ac.jp (M.I.); ishihara-im@med.kawasaki-m.ac.jp (K.I.); 3Department of Nephrology and Hypertension, Kawasaki Medical School, Kurashiki 701-0192, Japan; HajimeNagasu@kms-ndh.com (H.N.); kashinao@med.kawasaki-m.ac.jp (N.K.); 4Department of Biomedical Sciences and Comprehensive Care, Indiana University School of Dentistry, Indianapolis, IN 46202, USA; uekiy@iu.edu; 5Indiana Center for Musculoskeletal Health, Indiana University School of Medicine, Indianapolis, IN 46202, USA

**Keywords:** Src homology 3 domain-binding protein 2, systemic lupus erythematosus, lupus mouse model, dendritic cells

## Abstract

Background: The adaptor protein Src homology 3 domain-binding protein 2 (SH3BP2) is widely expressed in immune cells. It controls intracellular signaling pathways. The present study was undertaken to investigate the role of SH3BP2 in a murine systemic lupus erythematosus model. Methods: For the lupus model, we used *Fas^lpr/lpr^* mice. Clinical and immunological phenotypes were compared between *Fas^lpr/lpr^* and SH3BP2-deficient *Fas^lpr/lpr^* mice. Splenomegaly and renal involvement were assessed. Lymphocyte subsets in the spleen were analyzed by flow cytometry. To examine the role of SH3BP2 in specific cells, B cell-specific SH3BP2-deficient lupus mice were analyzed; T cells and bone marrow-derived dendritic cells and macrophages were analyzed in vitro. Results: SH3BP2 deficiency significantly reduced lupus-like phenotypes, presented as splenomegaly, renal involvement, elevated serum anti-dsDNA antibody, and increased splenic B220^+^CD4^−^CD8^−^ T cells. Notably, SH3BP2 deficiency in B cells did not rescue the lupus-like phenotypes. Furthermore, SH3BP2 deficiency did not substantially affect the characteristics of T cells and macrophages in vitro. Interestingly, SH3BP2 deficiency suppressed the differentiation of dendritic cells in vitro and reduced the number of dendritic cells in the spleen of the lupus-prone mice. Conclusions: SH3BP2 deficiency ameliorated lupus-like manifestations. Modulating SH3BP2 expression could thus provide a novel therapeutic approach to autoimmune diseases.

## 1. Introduction

Systemic lupus erythematosus (SLE) is an autoimmune disease influenced by genetic and environmental factors and is characterized by dysregulated immune responses such as loss of self-tolerance to cellular antigens and autoantibody production [1]. Autoantibody production causes immune complex formation, resulting in local and systemic inflammation and organ damage [1]. In addition to the involvement of the acquired immune system, the innate immune system also contributes to the induction and progression of SLE [2,3]. The clinical features of SLE have been recapitulated in several animal models [4,5,6,7], one of which is MRL-*Fas^lpr/lpr^* mice, which carry a loss-of-function mutation in the death-receptor, *Fas/CD95* [6]. The *Fas* mutation results in decreased Fas-mediated apoptosis of autoreactive lymphocytes and subsequent accumulation of these cells [8]. The lupus-prone mice develop massive lymphoproliferation and multiple-organ damage associated with increased autoreactive lymphocytes and autoantibodies, such as the anti-double-stranded DNA (dsDNA) antibody and rheumatoid factor (RF) [9]. 

Src homology 3 domain-binding protein 2 (SH3BP2, also known as 3BP2) is an adapter protein expressed primarily in immune cells, such as myeloid cells [10,11], B cells [12,13], and T cells [14]. SH3BP2 regulates immune-cell functions by interacting with intracellular signaling proteins, including Syk, PLCγ, Vav, and Src [15,16,17,18,19,20]. *SH3BP2* mutations are identified as being responsible for the genetic disorder cherubism (OMIM #118400) characterized by jaw-bone destruction [21]. In cherubism, *SH3BP2* mutations, such as Pro418Arg mutation, hyperactivate the downstream signaling in a gain-of-function manner via increased SH3BP2 protein [22]. *SH3BP2* gain-of-function mutations cause increased activation of macrophages and osteoclasts [10,23]. On the other hand, SH3BP2 deficiency has been reported to impair B-cell proliferation in response to BCR ligation [12,13]. Also, we have previously reported that SH3BP2 deficiency suppresses antibody production against type II collagen and markedly prevents the development of arthritis in a collagen-induced arthritis model [24]. However, it has not been determined whether and how the SH3BP2 deficiency suppresses the development of autoimmune diseases other than rheumatoid arthritis.

In this study, we investigated the involvement of SH3BP2 in SLE pathophysiology using SH3BP2-deficient mice and *Fas^lpr/lpr^* lupus-prone mice. We explored differential phenotypes in immune cell subpopulations isolated from SH3BP2-deficient mice to elucidate potential mechanisms driving SH3BP2-regulated autoimmune responses. 

## 2. Results

### 2.1. SH3BP2 Deficiency Improves Splenomegaly and Glomerular Proliferative Changes in Lupus-Prone Mice

SH3BP2 mRNA and protein are ubiquitously expressed in immune cells. The expression levels were examined in individual immune subsets. Quantitative PCR and immunoblot analyses revealed that SH3BP2 mRNA and protein are expressed in T cells, B cells, macrophages, and dendritic cells. Moreover, the expression levels are relatively higher in B cells, macrophages, and dendritic cells compared to T cells (Figure A1a,b, Appendix A). 

To assess the involvement of SH3BP2 in SLE pathogenesis, we generated SH3BP2-deficient *Fas^lpr/lpr^* mice. We first confirmed that SH3BP2 protein was deleted in the tissues of the SH3BP2-deficient mice (Figure 1a); SH3BP2 protein levels in the tissues were not altered by the *Fas^lpr/lpr^* mutation (Figure 1a). We observed *Sh3bp2^+/+^*, *Sh3bp2*^Δ*/*Δ^, *Fas^lpr/lpr^*, and *Sh3bp2*^Δ*/*Δ^*Fas^lpr/lpr^* mice until the age of 35 weeks. Two *Fas^lpr/lpr^* mice died before the age of 35 weeks, thus we analyzed *Sh3bp2^+/+^* (*n* = 10), *Sh3bp2*^Δ*/*Δ^ (*n* = 10), *Fas^lpr/lpr^* (*n* = 14), and *Sh3bp2*^Δ*/*Δ^*Fas^lpr/lpr^* mice (*n* = 14). We found that SH3BP2 deficiency significantly improved splenomegaly in the *Fas^lpr/lpr^* mice (Figure 1b,c). 

Given that renal involvement is a common feature in *Fas^lpr/lpr^* mice [7], we analyzed urinary protein levels and histological changes of the kidney. Urinary protein was not increased in the *Fas^lpr/lpr^* mice at this age (Figure A2a, Appendix A), whereas proliferative glomerular lesions were already present in the kidney of *Fas^lpr/lpr^* mice, and the proliferative change was retrieved in *Sh3bp2*^Δ*/*Δ^*Fas^lpr/lpr^* mice (Figure 1d,e). Glomerular sclerosis scores were not substantially aggravated in both *Fas^lpr/lpr^* and *Sh3bp2*^Δ*/*Δ^*Fas^lpr/lpr^* mice, and the scores were comparable between the genotypes (Figure A2b, Appendix A). These findings indicated that SH3BP2 deficiency ameliorated the early glomerular damage of lupus-prone mice.

### 2.2. SH3BP2 Deficiency Suppresses Autoantibody Production

Autoantibody production is a critical process associated with the development of organ damage in patients with SLE and lupus-prone mice [5,25,26]. We thus determined serum concentrations of the anti-dsDNA antibody (IgG) and RF (IgM). *Fas^lpr/lpr^* mice exhibited elevated levels of serum anti-dsDNA antibody and RF, whereas *Sh3bp2*^Δ*/*Δ^*Fas^lpr/lpr^* mice showed significantly suppressed production of these antibodies (Figure 2a,b). Additionally, we examined the serum immunoglobulin levels. *Sh3bp2*^Δ*/*Δ^*Fas^lpr/lpr^* mice exhibited significantly reduced production of IgM, and IgG2b, and total immunoglobulins compared to *Fas^lpr/lpr^* mice (Figure A3a,b, Appendix A). The ratios of anti-dsDNA antibody (IgG) per total IgG and RF (IgM) per IgM were significantly decreased in *Sh3bp2*^Δ*/*Δ^*Fas^lpr/lpr^* mice compared to those in *Fas^lpr/lpr^* mice (Figure A3c,d, Appendix A). These data indicate that SH3BP2 deficiency suppressed immunoglobulin production and that the suppression is likely to be predominantly in autoantibody levels.

### 2.3. SH3BP2 Deficiency Blunts the Aberrant Accumulation of CD3^+^B220^+^CD4^-^CD8^-^ Double-Negative T (DNT) Cells

*Fas^lpr/lpr^* lupus-prone mice are known to exhibit an aberrant pattern of T-cell subsets. One of the prominent subsets is B220^+^CD4^-^CD8^-^ T cells, referred to as DNT cells [27]. Accumulation of these cells is caused by impaired Fas-mediated apoptosis of autoreactive lymphocytes [28]. Also, *Fas^lpr/lpr^* mice have been reported to exhibit decreased naïve T cells and increased effector and activated T cells than those of wild-type mice [29]. We thus examined if SH3BP2 deficiency affects DNT cells in *Fas^lpr/lpr^* mice. The proportion of DNT cells in the spleen of *Fas^lpr/lpr^* mice was significantly elevated compared to that in *Sh3bp2^+/+^* mice (Figure 3a). Interestingly, the aberrant accumulation of DNT cells in the spleen was improved in the *Sh3bp2*^Δ*/*Δ^*Fas^lpr/lpr^* mice (Figure 3a,b). 

Next, to examine the activation status of the CD4^+^ and CD8^+^ T cells, we determined the early activation marker CD69 expression. Activated CD69^+^CD4^+^ T cells were increased in *Fas^lpr/lpr^* mice as previously reported [30], but SH3BP2 deficiency significantly suppressed the activation of CD4^+^ T cells in the spleen (Figure 3c). These findings suggest that the aberrant accumulation of autoreactive and activated T cells in lupus-prone mice was partly abrogated by SH3BP2 deficiency.

### 2.4. B Cell-Specific SH3BP2 Deficiency Does Not Improve the Clinical and Immunological Phenotypes

We hypothesized that SH3BP2 deficiency in B cells suppresses the production of pathogenic antibodies and subsequently ameliorates lupus-like phenotypes in *Fas^lpr/lpr^* mice. To this end, we generated conditional B cell-specific SH3BP2-deficient mice and examined if the deletion of SH3BP2 only in B cells improves the lupus-like phenotypes, as observed in the systemic knockout mice. 

We confirmed that SH3BP2 protein was deleted specifically in the B cells of the *Cd19^Cre/+^Sh3bp2^fl/fl^Fas^lpr/lpr^* mice (Figure A4a, Appendix A). We subsequently examined the phenotypes of the conditional knockout mice. Unexpectedly, B cell-specific deletion of SH3BP2 did not improve splenomegaly (Figure A4b,c, Appendix A). Also, B cell-specific SH3BP2 deletion did not ameliorate increased anti-dsDNA antibody levels and aberrant accumulation of DNT cells (Figure A4d,e, Appendix A). These findings indicate that SH3BP2 in B cells does not play a critical role in autoantibody production and development of the lupus-like phenotypes of *Fas^lpr/lpr^* mice.

### 2.5. SH3BP2 Deficiency Does Not Directly Affect the Activation of CD4^+^ T Cells

To examine whether SH3BP2 deficiency could affect CD4^+^ T cell activation, CD4^+^ T cells isolated from the spleen were stimulated with immobilized anti-CD3/anti-CD28. We found that there was no difference in the proportion of CD4^+^ T cells expressing CD69, CD25, CD44, and CD62L between control *Sh3bp2^+/+^* and *Sh3bp2*^Δ*/*Δ^ mice (Figure A5a, Appendix A). CD4^+^ T cell proliferation, assessed as carboxyfluorescein succinimidyl ester (CFSE) division, was not affected by SH3BP2 deficiency (Figure A5b, Appendix A). These findings indicate that SH3BP2 deficiency is unlikely to directly affect CD4^+^ T cell activation.

### 2.6. SH3BP2 Deficiency Modulates Optimal In Vitro Differentiation of Bone Marrow-Derived Dendritic Cells (DCs)

Since SH3BP2 has been reported to regulate the functions of myeloid cells [10,24], we examined if SH3BP2 deficiency could modulate the activation of macrophages. We found that SH3BP2 deficiency did not alter the expression levels of *Cd80* and *Cd86* mRNA, regardless of *Fas^lpr/lpr^* mutation (Figure A6a,b, Appendix A). Also, SH3BP2 deficiency did not affect the responsiveness of macrophages to TLR ligands, represented as *Tnf* mRNA expression (Figure A6c, Appendix A).

Next, we explored the differentiation of bone marrow cells into DCs. Interestingly, the differentiation into DCs, as determined by CD11c expression, was suppressed in both *Sh3bp2*^Δ*/*Δ^ and *Sh3bp2*^Δ*/*Δ^*Fas^lpr/lpr^* cells, compared to that in *Sh3bp2^+/+^* and *Fas^lpr/lpr^* cells, respectively (Figure 4a). The expression of MHC class II, CD80, CD86, and ICAM1 (CD54) gated on CD11c^+^ cells was diminished by the SH3BP2 deficiency in some of the CD11c^+^ cells (Figure 4a). These data suggest that SH3BP2 deficiency suppressed the differentiation of myeloid precursor cells into DCs.

Furthermore, we explored the effect of SH3BP2 on splenic DCs of the lupus-prone mice. The number of CD11c^+^MHC class II^+^ cells in the spleen was significantly reduced in the *Sh3bp2*^Δ*/*Δ^*Fas^lpr/lpr^* mice compared to that in *Fas^lpr/lpr^* mice (Figure 4b). This finding suggests that SH3BP2 could regulate lupus-like phenotypes via modulating the differentiation of DCs in the lupus-prone mice.

## 3. Discussion

SH3BP2 protein is ubiquitously expressed in various immune cells and plays critical roles in immune responses [31]. We have previously reported that in a collagen-induced arthritis model, SH3BP2 deficiency suppresses the development of arthritis by reducing the production of anti-type II collagen antibody [24], which was the first evidence showing the involvement of SH3BP2 in an autoimmune disease model. The present study demonstrates that SH3BP2 deletion significantly improves the clinical and immunological phenotypes of lupus-prone *Fas^lpr/lpr^* mice. 

We noted several distinct features of SH3BP2 deficiency in lupus-prone mice. First, germline deletion of SH3BP2 attenuated splenomegaly, glomerular proliferative changes, autoantibody production, and the accumulation of DNT cells. Second, these improved phenotypes were not rescued by B cell-specific deletion of SH3BP2. Third, SH3BP2 deficiency did not directly affect the in vitro activation of CD4^+^ T cells and macrophages. Lastly, SH3BP2 deficiency suppressed the differentiation of myeloid cells into DCs in vitro and reduced the number of splenic DCs in the lupus-prone mice. Collectively, SH3BP2 deficiency improved the lupus-like phenotypes of *Fas^lpr/lpr^* mice, which is likely caused by SH3BP2-mediated alteration in the myeloid cell differentiation into DCs.

We initially assumed that SH3BP2 in B cells would be essential for its regulatory effect on the pathogenesis of lupus because SH3BP2 has been reported to regulate B-cell functions [12,13]. Previous studies have shown that SH3BP2 deletion diminished the intracellular signaling activation, proliferation [12], and in-vivo thymus-independent type 2 antigen response [13]. Additionally, we also have reported that SH3BP2 deficiency suppresses pathogenic antibody production in a murine arthritis model [24]. However, the deletion of SH3BP2 in B cells, unexpectedly, did not retrieve the lupus phenotypes, including aberrant autoantibodies production. The findings indicate that SH3BP2 in B cells has limited effects on the phenotypes of lupus-prone mice. Also, these results suggest that the functional importance of SH3BP2-mediated cellular functions in B cells might vary depending on physiological or pathological conditions. 

Our present study has also shown that SH3BP2 deficiency does not influence in vitro activation and proliferation of CD4^+^ T cells. These findings are consistent with previous studies [12,13], in which SH3BP2 deficiency in T cells does not affect proliferation, IL-2 production, and intracellular signaling activation. These findings suggest that SH3BP2 is less likely to directly modulate cellular functions in CD4^+^ T cells in lupus-prone mice. 

Interestingly, SH3BP2 deficiency suppressed in vitro differentiation of DCs from myeloid precursor cells. DCs are the main regulators of innate and adaptive immune responses and play critical roles for initiation, amplification, and perpetuation of various diseases. Multiple studies in humans and mouse models have shown the involvement of DCs in the pathogenesis of lupus [32]. DCs from patients with SLE reportedly display significant phenotypical changes that promote aberrant T cell function [33,34]. Notably, the depletion of DCs in lupus-prone MRL/lpr mice reduces T cell expansion and autoantibody production [35], suggesting that DCs could be potential target cells in the treatment of lupus. In the present study, we demonstrated, for the first time, that the differentiation of DCs is suppressed by the deletion of SH3BP2. We thus propose that SH3BP2-mediated alteration of the differentiation into DCs is a possible mechanism for the improvement of lupus-like phenotypes in *Sh3bp2*^Δ*/*Δ^*Fas^lpr/lpr^* mice. Conditional knockout of SH3BP2 in DCs would be necessary to draw a definitive conclusion that proves the direct effects of SH3BP2 deficiency in DCs.

We have previously reported that SH3BP2 P416R gain-of-function mutation, which results in excessive SH3BP2 protein expression, ameliorates clinical and immunological phenotypes of *Fas^lpr/lpr^* lupus-prone mice [36]. In the study, we have shown that SH3BP2 gain-of-function mutation suppressed the accumulation of DNT cells, and increased expression of caspase-3 protein and *Tnf* mRNA in lymph nodes. Based on these findings, we considered that increased TNF expression induces the apoptosis of autoreactive lymphocytes independently of Fas-mediated apoptosis, which is, at least in part, responsible for ameliorating the phenotypes of *Fas^lpr/lpr^* mice. Interestingly, our present study has shown that SH3BP2 deficiency also ameliorates the lupus-like phenotypes of *Fas^lpr/lpr^* mice. The seemingly contradictory findings may suggest that different cells are affected by the excess or loss of the SH3BP2 protein. For instance, SH3BP2 gain-of-function significantly augments macrophage inflammatory responses [10,11,22,37], whereas SH3BP2 deficiency do not impact macrophage cellular responses [11,22]. As reported in our previous study [36], increased proapoptotic events mediated by SH3BP2 gain-of-function are considered to ameliorate lupus phenotypes in the *Fas^lpr/lpr^* lupus model. Of note, such events, including the expression of *Tnf* mRNA, were not observed in macrophages from *Sh3bp2*^Δ*/*Δ^*Fas^lpr/lpr^* mice in the current study (Figure A6c, Appendix A). These findings suggest that SH3BP2 deficiency has a minimal effect on the cellular functions of macrophages; therefore, probably, the role of SH3BP2 is interchangeable with that of other proteins in macrophages. On the other hand, with respect to DCs, SH3BP2 deficiency plays an essential role in their differentiation, as shown in the current study. Thus, we propose that the differences in the necessity and redundancy of SH3BP2 protein between macrophages and DCs are behind the ameliorated organ damage in both gain- and loss-of-function mice. To dissect the detailed mechanisms, further analyses using cell-specific gain- or loss-of-function mice will be required.

Improved lupus-like phenotypes in SH3BP2-deficient mice led us to consider that the modulation of SH3BP2 expression would be a potential therapeutic approach for SLE. Suppression of SH3BP2 protein levels might improve immunological abnormalities and organ involvement in systemic autoimmune diseases, including SLE. This concept is also supported by our previous findings that SH3BP2 deficiency suppresses the induction of collagen-induced arthritis [24]. From this standpoint, tankyrase can be a therapeutic target to modulate SH3BP2 protein expression. Tankyrase is a poly(ADP-ribose) polymerase that directly binds to SH3BP2 protein and subsequently mediates its degradation [22,38]. Increased tankyrase activity decreases SH3BP2 protein levels. The modulation of tankyrase activity would have therapeutic effects on autoimmune diseases, including SLE. Therefore, further investigations into the cell-specific function and regulation of SH3BP2/tankyrase and the development of a cell-specific targeting approach will be required. 

From the clinical standpoint, to understand whether the expression levels of SH3BP2 in immune cells correlate with disease activity and whether SH3BP2 can be a potential prognostic marker is of importance. Although the expression levels of SH3BP2 have not yet been investigated in patients with SLE, the expression patterns of Syk have been analyzed [39]; of note, Syk is the main binding partner of SH3BP2, and SH3BP2 promotes the efficient activation of Syk [10,20]. Interestingly a previous study reported that the activation of Syk in B cells was increased in patients with SLE (*versus* healthy controls), and correlated with disease activity [39]. Therefore, these findings suggest the possible association between the expression levels of SH3BP2 and disease activity. Clinical studies focusing on SH3BP2 would clarify the potential utility of SH3BP2 as a biomarker of SLE. 

This study provides a novel insight into the pathological role of SH3BP2 in the immunological mechanisms in SLE. Our current findings warrant further research on developing a novel therapeutic approach to autoimmune diseases by modulating the expression of SH3BP2.

## 4. Materials and Methods

### 4.1. Mice

B6. MRL*-Fas^lpr/lpr^* mice (#000482, referred to as *Fas^lpr/lpr^* mice) and *Cd19-Cre* mice (#006785) were obtained from Jackson Laboratory (Bar Harbor, ME, USA). *Sh3bp2*-floxed mice have been generated by inGenious Targeting Laboratory, as reported previously [40]. In the *Sh3bp2*-floxed mice, *Sh3bp2* exon3 is flanked by loxP sites. To generate B cell-specific SH3BP2-deficient mice, the *Sh3bp2*-floxed mice were crossed with the *Cd19-Cre* mice. To generate SH3BP2 systemic knockout mice, the *Sh3bp2*-floxed mice were crossed with *EIIa-Cre* mice (#003724, Jackson Laboratory), in which Cre recombinase is expressed in the germ cells [41]. Elimination of the *EIIa-Cre* allele was achieved by crossing with mice without the *EIIa-Cre* allele, resulting in SH3BP2 systemic knockout (*Sh3bp2*^Δ*/*Δ^) mice. All mouse studies were performed using male and female mice of the C57BL/6J background. All mutant mice were maintained in the animal facility of Kawasaki Medical School (Okayama, Japan), and were housed in a group (2–5 mice/cage) and maintained at 22 °C under a 12h:12h light/dark cycle with free access to water and standard laboratory food (MF diet, Oriental Yeast Co., Tokyo, Japan). All animal experiments were approved by the Safety Committee for Recombinant DNA Experiments (Nos.14-32 (7 January 2015), 15-24 (9 October 2015), and 20-28 (2 September 2020)) and the Institutional Animal Care and Use Committee of Kawasaki Medical School (Nos. 17-109 (7 December 2017), 18-105 (13 November 2018), 18-131 (4 February 2019), and 18-132 (4 February 2019)). All experimental procedures were conducted by the institutional and National Institute of Health guidelines for the humane use of animals.

### 4.2. Animal Study: Analysis of the Lupus-Prone Mice

To generate SH3BP2-deficient lupus-prone mice, *Fas^lpr/lpr^* mice were crossed with *Sh3bp2*^Δ*/*Δ^ mice (C57BL/6J background) to yield double-mutant mice. *Sh3bp2^+/+^* (*n* = 10), *Sh3bp2*^Δ*/*Δ^ (*n* = 10), *Fas^lpr/lpr^* (*n* = 16), and *Sh3bp2*^Δ*/*Δ^*Fas^lpr/lpr^* (*n* = 14) mice were monitored. Two *Fas^lpr/lpr^* mice were dead before 35 weeks of age; thus, 14 *Fas^lpr/lpr^* mice were analyzed. At the end of the observation period, samples of blood, spleen, and kidney were collected and used for subsequent analyses. Additional serum samples were collected at the same age from mice of another series of observational study, yielding increased sample numbers for serum assays; *Sh3bp2^+/+^* (*n* = 15), *Sh3bp2*^Δ*/*Δ^ (*n* = 16), *Fas^lpr/lpr^* (*n* = 18), and *Sh3bp2*^Δ*/*Δ^*Fas^lpr/lpr^* (*n* = 19).

To generate B cell-specific SH3BP2-deficient lupus-prone mice, *Fas^lpr/lpr^* mice were crossed with *Cd19^Cre/+^Sh3bp2^fl/fl^* mice (C57BL/6J background). *Cd19^Cre/+^Sh3bp2^fl/fl^Fas^lpr/lpr^* (*n* = 14), and *Cd19^+/+^Sh3bp2^fl/fl^Fas^lpr/lpr^* (*n* = 14) were monitored. One *Cd19^Cre/+^Sh3bp2^fl/fl^Fas^lpr/lpr^* mouse was dead before 35 weeks of age; thus, 13 *Cd19^Cre/+^Sh3bp2^fl/fl^Fas^lpr/lpr^* mice were analyzed. At the end of the observation period, samples of blood, spleen, and kidney were collected and used for subsequent analyses.

Both male and female mice were included in the mouse studies; no sex-differences were observed in the clinical and immunological phenotypes.

### 4.3. Western Blot Analysis

Protein expression in the lymph nodes, spleen, and B cells was determined by western blot, as described previously [36,42]. For the preparation of protein samples, tissues were harvested from 35-week-old mice immediately after euthanasia and soaked in the RIPA lysis buffer (Sigma-Aldrich, St. Lois, MO, USA) containing a protease inhibitor cocktail (P8340, Sigma-Aldrich). The tissues were minced using homogenizers. After centrifugation (15,000× *g* at 4 °C for 20 min), supernatants were collected, and protein concentrations were determined using a BCA protein assay kit (Thermo Fisher Scientific, Waltham, MA, USA). Protein samples were resolved by sodium dodecyl sulfate-polyacrylamide gel electrophoresis and transferred to nitrocellulose membranes. After blocking with 5% skim milk in Tris-buffered saline with 0.1% Tween-20, the membranes were incubated with the indicated primary antibodies, followed by incubation with the appropriate horseradish peroxidase (HRP)-conjugated species-specific secondary antibodies. The bands were detected using SuperSignal West Dura or Femto chemiluminescent substrate (Thermo Fisher Scientific) and visualized using an ImageQuant LAS-4000 (GE Healthcare, Little Chalfont, UK). Actin was used as a loading control to normalize the amount of protein. The antibodies used in this study were as follows: mouse anti-SH3BP2 monoclonal antibody (clone1E9; Abnova, Taipei City, Taiwan) and rabbit anti-Actin polyclonal antibody (A2066; Sigma-Aldrich).

### 4.4. Isolation of Mouse Primary T Cells and B Cells

Mouse CD4^+^ and CD8^+^ T cells and B cells in spleen were isolated using a CD4 isolation kit II (Miltenyi Biotec, Bergisch Gladbach, Germany), CD8a isolation kit II (Miltenyi Biotec), and Pan-B cell isolation kit II (Miltenyi Biotec) by negative selection, respectively. A purity rate of >96.6% for isolated CD4^+^ and CD8^+^ T cells and B cells was confirmed by flow cytometry [43]. 

### 4.5. Urinary Protein Assessment

Spot urine samples were collected from 35-week-old mice. Proteinuria was individually evaluated by using urine test strips (Albustix; Siemens Healthineers, Tokyo, Japan), as described [36]. Levels of proteinuria were semi-quantitatively graded from ± to 3+ (±, <30 mg/dL; 1+, 30–99 mg/dL; 2+, 100–299 mg/dL; 3+, 300< mg/dL). 

### 4.6. Histopathologic Assessment of Kidney

The kidneys were fixed in 4% paraformaldehyde for 2 days and then embedded in paraffin. Kidney sections (2 µm) were stained with periodic acid-Schiff (PAS). Glomerular pathology was assessed on 10 glomerular cross-sections per kidney, and the number of nuclei per glomerulus was calculated to determine glomerular cell hyperproliferation [44]. In each group, the percentage of those exhibiting segmental sclerosis was determined and expressed as glomerular sclerosis index, as reported previously with some modification [45]. Glomerular sclerosis was graded semiquantitatively in each glomerulus using a scale of 0 to 4: 0, no glomerular sclerosis; 1, mesangial matrix expansion; 2, segmental glomerular sclerosis in <25% of glomeruli; 3, segmental glomerular sclerosis in 25% to 50% of glomeruli; and 4, segmental glomerular sclerosis in >50% of glomeruli. The mean score per glomerulus was determined for each mouse.

### 4.7. Enzyme-Linked Immunosorbent Assay (ELISA) for Serum Anti-dsDNA Antibody and IgM-RF

Anti-dsDNA antibody (IgG) and IgM-RF levels in serum samples were measured by using ELISA kits (Shibayagi, Gumma, Japan) [36]. Diluted sera (anti-dsDNA 1:100 or IgM-RF 1:1000) were incubated on dsDNA- or RF-coated ELISA plates at 25 °C for 2 h. After washing, the plates were incubated with HRP-conjugated goat anti-mouse IgG and anti-mouse IgM, respectively, at 25 °C for 2 h. Tetramethylbenzidine was used for detection, and optical density at 450 nm (OD450) was measured using a microplate reader (Varioskan Flash; Thermo Fisher Scientific). Concentrations of anti-dsDNA antibodies (IgG) and IgM-RF were calculated and expressed as mU/ml. 

### 4.8. Serum Immunoglobulin Measurement

Concentrations of isotype-specific immunoglobulins in serum were measured with an ELISA kit, SBA Clonotyping System-C57BL/6-AP (Southern Biotech, Birmingham, AL, USA), as reported [36]. Each well in the 96-well plate was incubated with goat anti-mouse immunoglobulin (10 µg/mL) as a capture reagent at 4 °C overnight. Wells were blocked with 1% bovine serum albumin in phosphate-buffered saline at 25 °C for 1 h with gentle shaking. Diluted serum samples were added into the capture antibody-coated wells and incubated at 25 °C for 1 h with gentle shaking. The wells were then incubated with alkaline phosphatase (AP)-labeled detection antibodies at 25 °C for 1 h. After adding p-nitrophenyl phosphate substrate, optical densities were measured at 405 nm by a microplate reader (Varioskan Flash), and the concentrations of isotype-specific immunoglobulins were determined.

### 4.9. Flow Cytometry for Immune Cell Subsets

The subsets of immune cells in the spleen were analyzed with a flow cytometer (FACSCanto II; BD Biosciences, Franklin, NJ, USA), as described [36]. Briefly, we first prepared cell suspension by mincing spleen gently between two frosted slides. Red blood cell contamination was minimal and disappeared after treatment with red blood cell lysis buffer (eBioscience, San Diego, CA, USA). To block FcγR, single-cell suspensions were incubated with the rat anti-mouse CD16/CD32 antibody (2.4G2; BD Biosciences) on ice for 10 min before staining with the indicated monoclonal antibodies. The following monoclonal antibodies were used in this study: rat anti-mouse/human CD45R/B220 (RA3-6B2), rat anti-mouse CD4 (RM4-4), rat anti-mouse CD8a (53-6.7), Armenian hamster anti-mouse CD69 (H1.2F3, all from BioLegend, San Diego, CA, USA), and Armenian hamster anti-mouse/human CD3ε (145-2C11, eBioscience,); all antibodies were conjugated with fluorochrome. Dead cells were excluded by 7-aminoactinomycyn D (7-AAD; BioLegend) staining. In most samples, a minimum of 3 × 10^4^ events were evaluated, with all data analyzed using Flowjo software (version 10.6.1; BD Biosciences).

### 4.10. Functional Analyses of CD4^+^ T Cells

CD4^+^ T cells were isolated from the spleen of 12-week-old mice by negative selection using the CD4^+^ T Cell Isolation Kit (Miltenyi Biotec) according to the manufacturer’s instructions. Purified T cells were activated by plate-bound Armenian hamster anti-mouse CD3 antibody (145-2C11, 5 to 20 µg/mL; TONBO biosciences, San Diego, CA, USA) and/or golden Syrian hamster anti-mouse CD28 antibody (37.51, 2 µg/mL; TONBO biosciences) at 37 °C in RPMI1640 medium containing 10% heat-inactivated fetal bovine serum (FBS) and 0.05 mM 2-mercaptoethanol.

After the activation for 24 h, the cells were collected and stained for activation marker proteins on the cell surface using specific fluorophore-conjugated antibodies; rat anti-mouse CD4 (RM4-4), rat anti-mouse CD25 (PC61), rat anti-mouse CD62L (MEL-14), rat anti-mouse/human CD44 (IM7), and Armenian hamster anti-mouse CD69 (H1.2F3, all from BioLegend). For proliferation assessment, cells were stained with 1 µM carboxyfluorescein succinimidyl ester (CFSE; Dojindo laboratories, Kumamoto, Japan) in the dark at 25 °C for 10 min and were activated by plate-bound anti-CD3 antibody and/or anti-CD28 antibody for 72 h. Cells were collected at indicated time points and stained with a fluorophore-conjugated anti-CD4 antibody. All samples were analyzed in a flow cytometer (FACSCanto II) and Flowjo software.

### 4.11. Generation of Dendritic Cells (DCs) from Bone Marrow Cells

DCs were generated in vitro from bone marrow cells of 16-week-old mice, as described previously [36], with some modifications. Briefly, bone marrow cells were flushed from the tibia and femur of the mice; the collected cells were plated at a density of 1 × 10^6^/mL and cultured for 10 days in 5% CO_2_ at 37 °C in RPMI1640 medium containing 10% heat-inactivated FBS, 10 ng/mL recombinant mouse granulocyte-macrophage colony-stimulating factor (GM-CSF; PeproTech, Rocky Hill, NJ, USA), and 5 ng/mL mouse interleukin (IL)-4 (PeproTech). The yielded DCs were used in subsequent experiments.

### 4.12. Flow Cytometric Analyses of DCs

After the culture for 10 days with GM-CSF and IL-4, the DCs were collected and stained for marker proteins on the cell surface using specific fluorophore-conjugated antibodies; Armenian hamster anti-mouse CD11c (N418), Armenian hamster anti-mouse CD80 (16-10A1), rat anti-mouse CD86 (GL-1), rat anti-mouse CD54/ICAM1 (YN1/1.7.4, all from BioLegend), and rat anti-mouse MHC Class II (I-A/I-E; M5/114.15.2, eBioscience). Splenic cells isolated from 30-week-old mice were stained with Armenian hamster anti-mouse CD11c (N418, BioLegend) and rat anti-mouse MHC Class II (I-A/I-E; M5/114.15.2, eBioscience). Dead cells were excluded by 7-AAD (BioLegend) staining. All samples were analyzed in a flow cytometer (FACSCanto II) and Flowjo software.

### 4.13. Culture of Bone Marrow-Derived Macrophages

Isolation and culture of primary bone marrow cells were performed, as previously described [36,37]. Briefly, bone marrow cells were isolated from the long bones of 16-week-old female mice and cultured on Petri dishes for 2 h at 37 °C under 5% CO_2_. Non-adherent bone marrow cells were re-seeded on culture plates at a density of 1 × 10^6^ cells/mL and then incubated for 2 days in α-minimum essential medium (α-MEM) containing 10% heat-inactivated FBS and 25 ng/mL recombinant mouse macrophage colony-stimulating factor (M-CSF; PeproTech). After the 2-day pre-culture, the yielded bone marrow-derived macrophages were stimulated with Toll-like receptors ligands, such as lipopolysaccharide (LPS, Sigma-Aldrich), polyinosinic-polycytidylic acid sodium salt (Poly(I:C)), single-strand RNA (ssRNA), and CpG oligodeoxynucleotides (ODN, all from InvivoGen, San Diego, CA, USA), for 6 h in the presence of M-CSF. RNA samples were isolated from bone marrow-derived macrophages at the indicated time points and subjected to gene expression analysis.

### 4.14. Real-Time Quantitative Polymerase Chain Reaction (qPCR)

Total RNA was extracted from culture cells by using RNAiso Plus (Takara Bio, Shiga, Japan) and solubilized in RNase-free water as previously described [23,42]. cDNA was synthesized by using Prime Script RT reagent Kit (Takara Bio). qPCR reactions were performed by using TB Green PCR Master Mix (Takara Bio) with StepOne Plus System (Thermo Fisher Scientific). Gene expression levels relative to *Hprt* were calculated by ΔΔCt method and normalized to control samples indicated in each experiment. The qPCR analysis used following primers; 5′-tcgtctttcacaagtgtcttcag-3′ and 5′-ttgccagtagattcggtcttc-3′ for *Cd80*, 5′-gaagccgaatcagcctagc-3′ and 5′-cagcgttactatcccgctct-3′ for *Cd86*, 5′-catcttctcaaaattcgagtgaca-3′ and 5′-tgggagtagacaaggtacaaccc-3′ for *Tnf*, 5′-tcctcctcagaccgctttt-3′ and 5′-cctggttcatcatcgctaatc-3′ for *Hprt*, respectively. All qPCR reactions yielded products with single peak dissociation curves.

### 4.15. Statistical Analysis 

Individual values are presented as dots and the means or the means ± standard deviation. The distribution of the values was assessed by using the Kolmogorov-Smirnov test. Statistical analysis was performed by the two-tailed unpaired Student’s *t*-test or Mann-Whitney test to compare two groups and by one-way ANOVA (Tukey post-hoc test) or Kruskal-Wallis test (Dunn’s multiple comparison) to compare three or more groups using GraphPad Prism 5 (GraphPad Software, San Diego, CA, USA). *P* values less than 0.05 were considered statistically significant.

## Figures and Tables

**Figure 1 ijms-22-04169-f001:**
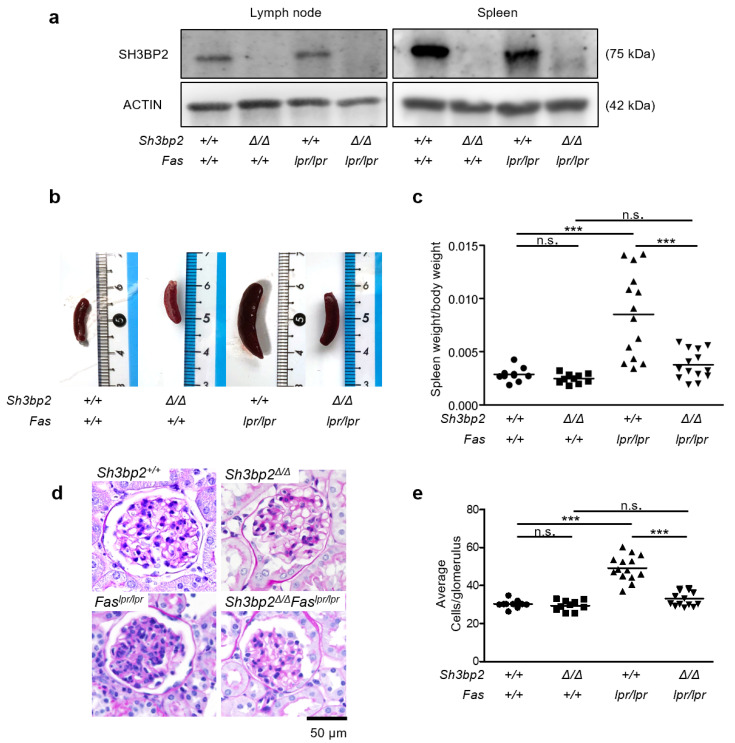
SH3BP2 deficiency improves splenomegaly and the proliferative changes in glomeruli of the *Fas^lpr/lpr^* lupus-prone mice. (**a**) Immunoblot analysis for SH3BP2. Protein samples were collected from lymph nodes and spleens of the indicated mice. SH3BP2 protein levels were determined by western blotting. Actin was used as the loading control. (**b**–**e**), *Sh3bp2^+/+^* (*n* = 10), *Sh3bp2*^Δ*/*Δ^ (*n* = 10), *Fas^lpr/lpr^* (*n* = 14), and *Sh3bp2*^Δ*/*Δ^*Fas^lpr/lpr^* mice (*n* = 14) were analyzed at the age of 35 weeks. (**b**) Representative images of the spleen. (**c**) Spleen weights per body weights were determined. (**d**) Representative images of periodic acid-schiff (PAS)-stained kidney sections. Original magnification, 400×. Bar, 50 μm. (**e**) Quantification of the number of cells in the glomeruli. At least 10 glomeruli per mouse were assessed. The numbers of nuclei in the glomerular cross-sections were counted, and the averages of the nuclei per glomerulus were determined. Each dot denotes an individual mouse, and horizontal lines represent the means. *** *p* < 0.001; n.s. = not significant.

**Figure 2 ijms-22-04169-f002:**
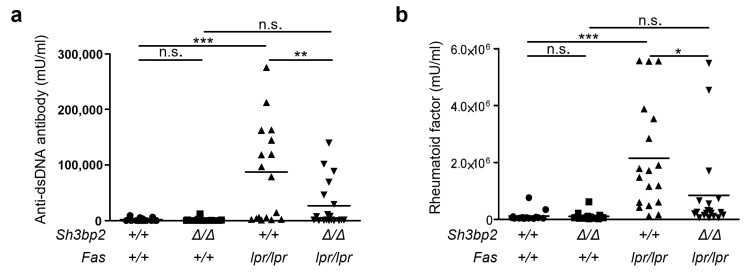
SH3BP2 deficiency suppresses elevated autoantibody production of *Fas^lpr/lpr^* lupus-prone mice. Serum samples were collected from *Sh3bp2^+/+^* (*n* = 15), *Sh3bp2*^Δ*/*Δ^ (*n* = 16), *Fas^lpr/lpr^* (*n* = 18), and *Sh3bp2*^Δ*/*Δ^*Fas^lpr/lpr^* (*n* = 19) mice at the age of 35 weeks. Levels of anti-dsDNA antibody (IgG) (**a**) and rheumatoid factor (IgM) (**b**) were determined by ELISA. Each dot denotes an individual mouse, and horizontal lines represent the means. * *p* < 0.05; ** *p* < 0.01; *** *p* < 0.001; n.s. = not significant.

**Figure 3 ijms-22-04169-f003:**
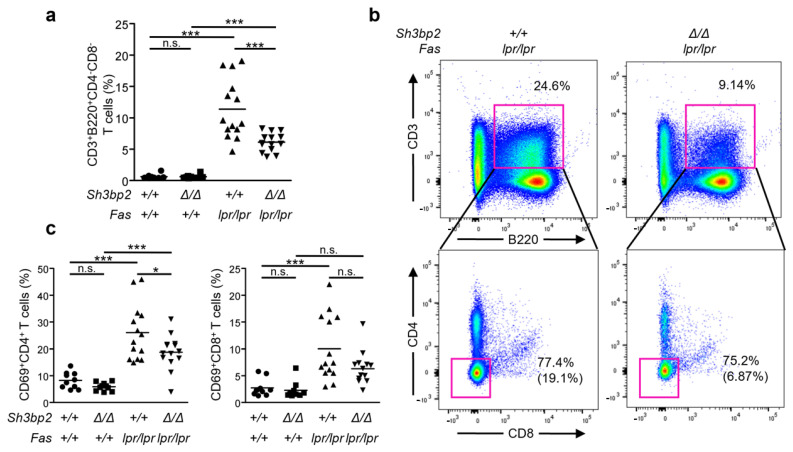
Aberrant accumulation of CD3^+^B220^+^CD4^-^CD8^-^ cells is improved in *Sh3bp2*^Δ*/*Δ^*Fas^lpr/lpr^* mice. Spleen cells were collected from *Sh3bp2^+/+^* (*n* = 10), *Sh3bp2*^Δ*/*Δ^ (*n* = 10), *Fas^lpr/lpr^* (*n* = 14), and *Sh3bp2*^Δ*/*Δ^*Fas^lpr/lpr^* (*n* = 14) mice at 35 weeks of age. T cell subsets were stained with fluorochrome-labeled antibodies against CD3, CD4, CD8, B220, and CD69. (**a**) The ratios of T cell subsets in the spleen were analyzed by flow cytometry. All the cells were gated on lymphocytes. (**b**) Representative flow cytometry plots of CD3^+^B220^+^CD4^-^CD8^-^ cells (double negative T (DNT) cells) cells. After gating CD3^+^B220^+^ cells, the ratios of CD4^-^CD8^-^ cells are defined as indicated by red rectangles. The numbers in parentheses indicate the percentages of DNT cells in total lymphocytes. (**c**) The ratios of CD69^+^CD4^+^ or CD8^+^ T cell subsets were determined. Each dot denotes an individual mouse, and horizontal lines represent the means. * *p* < 0.05; *** *p* < 0.001; n.s. = not significant.

**Figure 4 ijms-22-04169-f004:**
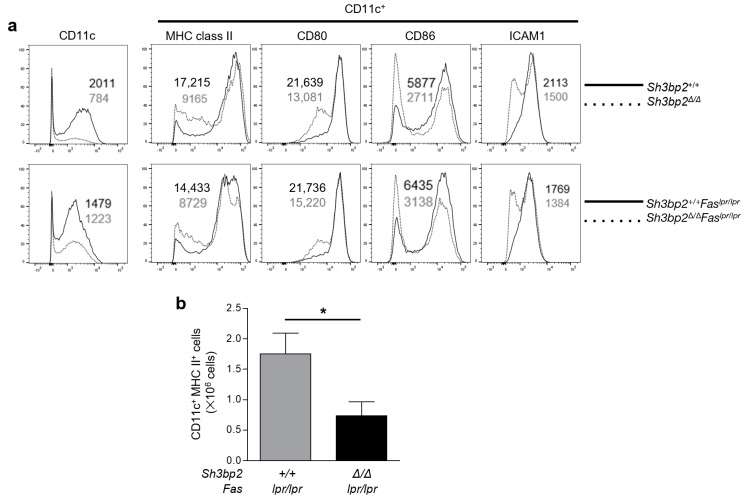
SH3BP2 deficiency suppresses the differentiation of dendritic cells. (**a**) Flow cytometric analyses of cell surface markers on dendritic cells. Bone marrow cells were isolated from 16-week-old *Sh3bp2^+/+^*, *Sh3bp2*^Δ*/*Δ^, *Fas^lpr/lpr^*, and *Sh3bp2*^Δ*/*Δ^*Fas^lpr/lpr^* mice. After a 10-day pre-culture with GM-CSF and IL-4, yielded bone marrow-derived dendritic cells were analyzed by flow cytometry. Representative histograms show the expression levels of CD11c gated on live cells and MHC class II, CD80, CD86, and ICAM1 (CD54) gated on CD11c^+^ cells. Data were obtained from at least two independent experiments. Values are the mean fluorescence intensities of the cells. (**b**) Flow cytometric analyses of the splenic cells. Spleen cells were collected from *Fas^lpr/lpr^* and *Sh3bp2*^Δ*/*Δ^*Fas^lpr/lpr^* (*n* = 3) mice at 30 weeks of age and stained with fluorochrome-labeled antibodies against CD11c and MHC class II. Total numbers of CD11c^+^MHC class II^+^ cells in the spleen were determined. Values are presented as the mean ± SD. * *p* < 0.05.

## Data Availability

Data is contained within the article.

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
