# Peer review of "SH3BP2 Deficiency Ameliorates Murine Systemic Lupus Erythematosus"

_ijms, 2021, doi:10.3390/ijms22084169_

Round 1

Reviewer 1 Report

I have no further comments for the authors.

Author Response

We appreciate your positive comment. Thank you again for your insightful comments that helped us to considerably improve our manuscript.

Reviewer 2 Report

I am happy to see the improvement of this manuscript with all the changes the authors made. However, without clear experimental evidences that can explain why gain-of-function and loss-of-function both result in ameliorated lupus disease, I just cannot feel very comfortable with the publication. I appreciate the efforts that the authors spent in improving the manuscript, but just as I mentioned in the first time revision, the solid evidence to show which cell type is critical will make this manuscript far more interesting.

Author Response

We appreciate your positive comments on our additional experiments and the revised manuscript. Thanks to your insightful comments, our manuscript has greatly improved. However, there are still unsolved issues as you pointed out. We also believe it would be fascinating to conduct experiments using cell-specific gain- or loss-of-function murine models. Such studies will clearly take substantial additional time to complete, and this is far beyond the revision deadline of 10 days. This said, we plan to perform the experiments in our follow-up study. Thank you again for taking the time and energy to help us improve the paper.

Reviewer 3 Report

In this article, Kawahara et al investigate the role of SH3BP2 in a murine systemic lupus erythematosus model. The Authors used Faslpr/lpr and SH3BP2-deficient Faslpr/lpr mice and they observed that SH3BP2 deficiency ameliorated lupus-like manifestations. In particular, by in vitro analysis SH3BP2 deficiency has been shown to suppress the differentiation of dendritic cells and to reduce the number of dendritic cells in the spleen of the lupus-prone mice.

This is a nice study that adds this study adds new insights into the role of SH3BP2 in lupus pathogenesis The research is novel and addresses an important health issue.

There are a few minor points that have to be addressed before the work could be considered for publication.

Minor points:

-the meaning of the text written in red is unclear

-it would be advisable to discuss whether the protein may play a role in patients and whether the expression levels of SH3BP2 in immune cells can correlate with disease activity, also suggesting a role as potential prognostic marker

- minor formatting errors in the references

Author Response

In this article, Kawahara et al investigate the role of SH3BP2 in a murine systemic lupus erythematosus model. The Authors used Faslpr/lpr and SH3BP2-deficient Faslpr/lpr mice and they observed that SH3BP2 deficiency ameliorated lupus-like manifestations. In particular, by in vitro analysis SH3BP2 deficiency has been shown to suppress the differentiation of dendritic cells and to reduce the number of dendritic cells in the spleen of the lupus-prone mice.

This is a nice study that adds this study adds new insights into the role of SH3BP2 in lupus pathogenesis The research is novel and addresses an important health issue.

Response: We appreciate your positive comments.

Minor points:

-the meaning of the text written in red is unclear 

Response: To clarify our statement, we have rephrased the description in the Discussion. Also, the discussion was edited by native English speakers. We hope the revision has made our statement clear and improved the readability.

-it would be advisable to discuss whether the protein may play a role in patients and whether the expression levels of SH3BP2 in immune cells can correlate with disease activity, also suggesting a role as potential prognostic marker

Response: We thank your constructive suggestion. Following your advice, we have added the following description in the Discussion.

From the clinical standpoint, to understand whether the expression levels of SH3BP2 in immune cells correlate with disease activity and whether SH3BP2 can be a potential prognostic marker is of importance. Although the expression levels of SH3BP2 have not yet been investigated in patients with SLE, the expression patterns of Syk have been analyzed [39]; of note, Syk is the main binding partner of SH3BP2, and SH3BP2 promotes the efficient activation of Syk [10,20]. Interestingly a previous study reported that the activation of Syk in B cells was increased in patients with SLE (versus healthy controls), and correlated with disease activity [39]. Therefore, these findings suggest the possible association between the expression levels of SH3BP2 and disease activity. Clinical studies focusing on SH3BP2 would clarify the potential utility of SH3BP2 as a biomarker of SLE.

- minor formatting errors in the references

Response: We thank you for pointing out. We have formatted the references.

Round 2

Reviewer 2 Report

Alright. Actually I don't like it to be published in current version because I believe it will turn to be much better with more data added. However, it looks like both the journal and authors want it to be published right now. 

Author Response

Thank you very much for your consideration. Our manuscript has greatly improved, thanks to your insightful comments.

This manuscript is a resubmission of an earlier submission. The following is a list of the peer review reports and author responses from that submission.

Round 1

Reviewer 1 Report

In the current manuscript, Kawahara et al. reports that SH3BP2 deficiency in B6.lpr mouse can reduce the lupus-like symptoms in this lupus-prone mouse model. The authors see the decrease of double negative T cells (CD4-CD8-) in spleen and lowered CD69 expression in CD4+ T cells. However, the B cell specific SH3BP2 deficiency cannot recapitulate the phenotype seen in SH3BP2 global knockout mice, indicating a B cell independent manner. The study is interesting but some important issues need to be addressed.

Major points:

  1. In one recent publication from the same research group - "Sh3bp2 Gain-Of-Function Mutation Ameliorates Lupus Phenotypes in B6.MRL- Fas lpr Mice", in the same B6.lpr mouse model, they reported the gain-of-function of SH3BP2 ameliorate lupus phenotypes. This is apparently opposite to each other and their discussion on this point is not good enough. If you compare the Sh3bp2+/+Fas+/+ bands in Fig 1a to the Sh3bp2 overexpression bands in previous publication, there is no difference. So in my opinion, it is possible that something wrong happened in the gene modification in these mice which lead to the weird results. Please examine carefully and provide reasonable explanations on this issue.
  2. Sample number is confusing and inconsistent in this paper. The authors always say that the number of mice in different groups are: n=10 in Sh3bp2+/+ and Sh3bp2-ko/ko groups, n=16 in B6.fas group and n=14 in Sh3bp2-ko/ko;fas group. However, the dot numbers in figures are not as described, sometimes even more dots than the claimed number. For example, in fig 2b, the dot number in B6.lpr group is 18 which is more than 16... 
  3. The mechanism is not figured out. The authors checked the B cell specific Sh3bp2 deficiency and find that it cannot recapitulate the phenotype witnessed in global ko mouse. I will encourage the authors to find out which cell type is the major player. The Sh3bp2 expression examinations in different cell types, RT-PCR and western, may help for any clues.

Minor points:

  1. In figure 1, please add kidney histopathology scores and proteinuria levels for 4 groups of mice.
  2. In figure 2, is the total Ig changed by Sh3bp2 deficiency? How about the ratio of anti-dsDNA versus total Ig. Because IgG is the major types of auto-ab, please also provide anti-dsDNA IgG data.
  3. Please move the B cell specific Sh3bp2 deficiency result into supplement because it's not the dependent mechanism.

Reviewer 2 Report

The study analyzed the role of SH3BP2 in a murine model of SLE. Clinical, immunological, splenic and renal involvement were assessed in Faslpr/lpr and SH3BP2 deficient Faslpr/lpr mice. SH3BP2 defieciency significantly reduced all the lupus manifestations in Faslpr/lpr mice.

The paper is interesting, however I have few comments for the authors:

  • In the Methods section it is stated that a to compare three or more group a one-way ANOVA with Turkey post-hoc test was used. Please report which test was used to assess the normal distribution of the data.
  • In figure 1, 2 and 3 the comparison between group 2 and group 4 was not reported. Please consider to report it.